# Urushiol-Based Benzoxazine Containing Sulfobetaine Groups for Sustainable Marine Antifouling Applications

**DOI:** 10.3390/polym15102383

**Published:** 2023-05-19

**Authors:** Jing Zhao, Jipeng Chen, Xiaoxiao Zheng, Qi Lin, Guocai Zheng, Yanlian Xu, Fengcai Lin

**Affiliations:** 1College of Chemistry and Materials, Fujian Normal University, Fuzhou 350007, China; 2Fujian Key Laboratory of Polymer Materials, Fujian Normal University, Fuzhou 350007, China; 3Fujian Provincial Key Laboratory of Advanced Oriented Chemical Engineering, Fujian Normal University, Fuzhou 350007, China; 4Fujian Engineering Research Center of New Chinese Lacquer Materials, Minjiang University, Fuzhou 350108, China

**Keywords:** benzoxazine, urushiol, zwitterionic polymer, marine antifouling

## Abstract

Benzoxazine resins are new thermosetting resins with excellent thermal stability, mechanical properties, and a flexible molecular design, demonstrating promise for applications in marine antifouling coatings. However, designing a multifunctional green benzoxazine resin-derived antifouling coating that combines resistance to biological protein adhesion, a high antibacterial rate, and low algal adhesion is still challenging. In this study, a high-performance coating with a low environmental impact was synthesized using urushiol-based benzoxazine containing tertiary amines as the precursor, and a sulfobetaine moiety into the benzoxazine group was introduced. This sulfobetaine-functionalized urushiol-based polybenzoxazine coating (poly(U−ea/sb)) was capable of clearly killing marine biofouling bacteria adhered to the coating surface and significantly resisting protein attachment. poly(U−ea/sb) exhibited an antibacterial rate of 99.99% against common Gram negative bacteria (e.g., *Escherichia coli* and *Vibrio alginolyticus*) and Gram positive bacteria (e.g., *Staphylococcus aureus* and *Bacillus* sp.), with >99% its algal inhibition activity, and it effectively prevented microbial adherence. Here, a dual-function crosslinkable zwitterionic polymer, which used an “offensive-defensive” tactic to improve the antifouling characteristics of the coating was presented. This simple, economic, and feasible strategy provides new ideas for the development of green marine antifouling coating materials with excellent performance.

## 1. Introduction

Marine biofouling refers to the adherence of marine organisms on the surfaces of marine equipment, leading to economic damage in terms of human marine activities [1,2]. Marine biofouling is a global issue with permanent detrimental consequences on machinery [3,4]. Currently, petroleum-based resin self-polishing antifouling coatings comprising cuprous oxide (Cu_2_O) and a powerful biocide (such as zinc thiopyridine, copper/zinc pyridine thione, etc.) are widely used at home and abroad [5,6]. Although these self-polishing antifouling coatings prevent marine organisms from fouling, the coatings exert serious toxic effects on non-target marine organisms and the environment [7,8]. Therefore, to solve the ecotoxicological problems caused by the use of toxic antifouling coatings, designing and building green, long-lasting, excellent performance, and environmentally friendly new antifouling coatings, such as hydrophilic antifouling coatings, is imperative [9].

Most of the hydrophilic coatings exhibit eco-friendly and marine eco-friendly properties, which are highly sought after by researchers [10]. Owing to their good biocompatibility, hydrophilicity, non-toxicity, and excellent resistance to protein adhesion, zwitterionic polymers were initially used in medical devices [11], cotton fabrics [12], and filter membranes [13], and later for marine antifouling applications [14,15]. Zwitterionic polymers comprise the same number of anions and cations, in which the quaternary ammonium salt (R_4_N^+^) directly penetrates the cell membrane surface and binds to the negative charge of the phospholipid bilayer on the cell membrane via electrostatic interactions, resulting in cell membrane destruction and bacterial necrosis. On the other hand, zwitterionic polymers exhibit strong electrostatic-induced hydration; as a result, a tight water layer is easily formed on the material’s surface, which in turn prevents the adhesion of fouling organisms effectively [16,17]. Chiang et al., prepared an antiprotein adhesion filtration membrane with a water flux recovery of 95.5% by grafting sulfobetaine methacrylate (SBMA) onto a polyvinylidene fluoride (PVDF) membrane surface [18]. Hibbs et al., prepared a series of polysulfone and polyacrylate-based amphoteric coatings on an epoxy resin primer aluminum substrate with 70% antibacterial and algal inhibition rates [19]. Liu prepared an innovative polymer brush on a resin matrix that can effectively inhibit the adhesion of red sponge larvae by one-step surface activation ATRP (SSI-ATRP) [20]. Zwitterionic polymers play a key role in hydrophilic antifouling coatings because of their good antifouling and environmental protection properties [21,22]. At the same time, under the “double carbon,” background, green, natural, easily extractable, and renewable natural products are selected as raw materials to prepare hydrophilic bio-based antifouling coatings. In addition, hydrophilic bio-based antifouling coatings are promising polymers because of their good hydrophilicity, cost-effectiveness, and commercial feasibility. However, most of the methods to synthesize zwitterionic materials coating materials involve direct polymerization and grafting with zwitterionic ions on the coating surface, which may involve complex chemical reactions and necessitate specific requirements on the coating surface; hence, commercialization is difficult. Therefore, designing and building zwitterionic bio-based coatings with excellent antifouling performance and in agreement with the modern “green environmental protection” marine antifouling concept is still challenging.

Meanwhile, raw lacquer is a type of lacquer juice harvested from lacquer trees, which can be used as a natural and durable coating. Urushiol (U) is the main film-forming substance in raw lacquer, with a content of 60–70% [23]. Urushiol-based polymers exhibit incomparable uniqueness of petroleum-based resin coatings in terms of their mechanical properties, substrate adhesion, antibacterial and aging resistance. By introducing other functional groups and compounding degradable components, the structure and properties of urushiol-based polymers can be regulated effectively, and a new type of a bio-based marine antifouling coating with excellent performance can be prepared [24]. Previously, our group has successfully synthesized a low-surface-energy marine antifouling coating, referred to as urushiol-based benzoxazine copper polymer (UBCP) with antifouling performance, using urushiol, octylamine, and a copper compound as the phenolic source, amine source, and catalyst, respectively [25]. The as-obtained UBCP was cured into films at room temperature, which exhibited strong substrate adhesion, a smooth and dense surface, and the controllable release of an effective copper ion at minimized concentrations; hence, excellent antifouling performance was imparted to UBCP. As a new type of a thermosetting resin, benzoxazine exhibits high temperature resistance, chemical stability, and high durability. In the cross-linking reaction, benzoxazine easily forms a chemical bond with the coating substrate, such as the ring-opening addition of the oxazine ring, in which active methylene reacts strongly with electrophilic groups; as most of the substrates contain electrophilic groups, the adhesion of benzoxazine to the substrate is good [26,27,28].

Based on the above premise, the introduction of sulfobetaine into benzoxazine monomers is assumed to not only improve the poor hydrophilicity of benzoxazine through anions and cations effectively, but also the amphoteric ionization of polymers is easily conducted in comparison to direct polymerization. Reactive polymers with an adjustable chemical structure can be prepared with a high zwitterionic function and good antifouling performance. In this study, urushiol-based benzoxazine (U−ea) is prepared by the Mannich condensation using urushiol, paraformaldehyde, and ethanolamine as the raw materials. Subsequently, sulfobetaine-functionalized benzoxazine (U−ea/sb) is prepared by the reaction of a tertiary amine on the oxazine ring with sulfonate. The synthesis is simple, with no by-product. The effects of zwitterionic groups on the surface wettability, thermal stability and physical and mechanical properties of the coating can be analyzed by testing the contact angle, droplet adhesion, thermogravimetric analysis and mechanical properties of the coating. In addition, the antifouling performance of poly(U−ea/sb) coatings is systematically investigated using typical Gram negative bacteria such as *Escherichia coli* (*E. coli*) and *Vibrio alginolyticus* (*V. alginolyticus*), Gram positive bacteria such as *Staphylococcus aureus* (*S. aureus*) and *Bacillus.* sp., and algal species such as *Nitzschia closterium* (*N. closterium*), *Phaeodactylum tricornutum* (*P. tricornutum*), and *Dicrateria zhanjiangensis* (*D. zhanjiangensis*). *Bovine serum albumin* (*BSA*) and *γ-globulin* are used to systematically evaluate the antifouling performance of the poly(U−ea/sb) coatings. In this paper, the development of antifouling coatings with significant hydrophilic and excellent antifouling properties is focused, and ideas are suggested for the preparation of new, efficient, eco-friendly, hydrophilic bio-based antifouling coatings with dual “offensive-defensive” antifouling mechanisms.

## 2. Materials and Methods

### 2.1. Materials

Chinese raw lacquer was purchased in Maoaba Town, Hubei Province, China. Urushiol (Figure 1) was extracted from Chinese raw lacquer using ethanol according to a previous method [29]. Figure 1 shows the chemical structure of urushiol. Paraformaldehyde (95 wt%), 1,4-dioxane, dichloromethane, xylene, tetrahydrofuran, and anhydrous sodium sulfate were purchased from Sinopharm Chemical Reagent Co., Ltd, Shanghai, China. Ethanolamine (99 wt%) and 1,3−propanesultone (98 wt%) were purchased from Shanghai Macklin Biochemical Co., Ltd. Deionized (DI) water was prepared in the laboratory using a WP-RO-10B water purifier system (Votel Water Treatment Equipment Co., Ltd., Sichuan, China). *BSA* and *γ-globulin* were purchased from Yuanye Biotechnology Co, Shanghai China. *E. coli*, *V. alginolyticus*, *S. aureus*, and *Bacillus.* sp. were purchased from Bioboway Biotechnology Co., Ltd., Beijing, China. *N. closterium*, *P. tricornutum*, and *D. zhejiangensis* were purchased from Guangyu Biotechnology Co., Ltd, Shanghai, China.

### 2.2. Characterization

^1^H NMR spectra were recorded on a Bruker 400 MHz NMR spectrometer (Bruker, Berlin, Germany) using CDCl_3_ as the solvent. Attenuated total reflectance Fourier transform infrared (ATR-FTIR) spectra were recorded on a Nicolet 5700 FTIR spectrometer (Thermo Fisher, Waltham, MA, USA). X-ray photoelectron spectroscopy (XPS) curves were recorded on a VG MultiLab 2000 XPS spectrometer (Thermo Fisher, Waltham, MA, USA). All binding energies were calibrated using the C1s peak at 284.8 eV. Water contact angles (WCA) were determined using a DSA 25 drop analyzer (Kruss, Hamburg, Germany) using 2-μL DI water droplets at ambient temperature. Five measurements were conducted for each sample, and the average value was taken as the CA of the coating. To analyze the glass transition temperature, differential scanning calorimetry (DSC) was performed on a DSC 3 instrument (Mettler-Toledo, Zurich, Switzerland) at a heating rate of 10 °C/min from −30 °C to 160 °C under nitrogen. The thermal stability of the samples was investigated by thermogravimetric analysis (TGA) on a TGA 3 system (Mettler-Toledo, Switzerland) under nitrogen at a flow rate of 50 mL/min and a ramp-up rate of 10 °C/min from 30 °C to 600 °C. FE−SEM (Hitachi SU8010, Tokyo, Japan) was employed to observe the adhesion of bacteria to glass slides, as well as on poly(U−ea) and the poly(U−ea/sb) coatings, at an accelerating voltage of 5 kV. A force tensiometer (K100MK2, Kruss, Hamburg, Germany) was used to measure adhesion force by using DI droplets (6 μL) at an immersion depth of 0.1 mm and an immersion time of 0.01 min at room temperature, measurements were conducted 5 times for each sample, and the mean and standard deviations were obtained. A TU-1810 UV-Vis spectrophotometer (Beijing, China) was employed for analysis, where 3 mL of a suspension was added in a cuvette, the full spectrum was scanned at wavelengths of 400–200 nm, and the scanning speed was set to medium. Absorbance was measured at the characteristic absorption wavelengths of *BSA* and *γ-globulin*. Conventional physical and mechanical properties: The hardness of the coating pencil was tested according to GB/T6739-2006 and ISO15187-2012 (pencil method for paint and varnish).

### 2.3. Synthesis of Urushiol-Based Benzoxazine Monomer (U−ea)

Figure 2 shows the preparation of U−ea. Briefly, paraformaldehyde (0.20 mol, 6.32 g), ethanolamine (ea) (0.1 mol, 6.10 g), and 1,4-dioxane (20 mL) were first loaded into a 250-mL three-neck round-bottom flask equipped with a thermometer, a reflux condenser, and a dropping funnel. Second, the mixture was stirred at room temperature for 40 min. Third, urushiol (U) (0.05 mol, 15.70 g) was dissolved in 1,4-dioxane (25 mL) and added to the flask within 20 min. Subsequently, the system was gradually warmed to 90 °C and maintained at this temperature under vigorous stirring for 6 h. After the system was cooled to room temperature, the solvent was removed by vacuum distillation to obtain crude U−ea. The crude product was dissolved in dichloromethane and washed several times with deionized water (DI) to purify the organic solution. Then, the organic solution was dried over anhydrous sodium sulfate overnight, and the solvent was removed by rotary evaporation, affording a reddish-brown viscous solution; this solution was the urushiol-based benzoxazine monomer, and it was dried under vacuum at room temperature for 24 h.

### 2.4. Introduction of Sulfobetaine Groups into Polybenzoxazine Chains

First, the U−ea monomer (0.01 mol, 4.01 g) was dissolved in 10 mL of tetrahydrofuran, and the mixture was added into a 100-mL three-neck round-bottom flask equipped with a thermometer, a reflux condenser, and a dropping funnel. After the addition of 1,3−propanesultone (0.01 mol, 1.22 g), the solution was reacted at 35 °C for 24 h. At the end of the reaction, the solvent was removed by vacuum distillation and dried under vacuum at room temperature for 24 h to obtain the sulfobetaine-functionalized benzoxazine monomer (U−ea/sb).

### 2.5. Fabrication of Poly(U−ea) and Poly(U−ea/sb) Coated Glass Slides and Tinplate Sheets

U−ea and U−ea/sb are applied dropwise to glass or tinplate sheets for surface wettability, stain resistance and mechanical properties of the coating. poly(U−ea) and poly(U−ea/sb) coated glass slides were prepared as follows: First, bare glass slides (BG, 2.0 cm × 2.0 cm) were cleaned by ultrasonication for 10 min sequentially using acetone, ethanol, and DI water, followed by drying under N_2_ to remove contaminants. Second, heat-cured urushiol-based polybenzoxazine (poly(U−ea)) and the poly(U−ea/sb) coating were prepared as follows: For reference, a certain amount of U−ea and dichloromethane (CH_2_Cl_2_) were loaded in a beaker and stirred until a homogeneous solution (40 wt%) was obtained. The solution was dropped onto the pretreated slides and first cured at room temperature for 1 h to allow the solvent to evaporate. Next, the obtained films were placed in an oven and heated for curing, followed by a programmed temperature increase from 100 °C to 160 °C at 20 °C/h to obtain dark brown poly(U−ea) coatings. poly(U−ea) and poly(U−ea/sb) coated tinplate sheets are prepared with reference to the methods described above.

## 3. Results

### 3.1. Synthesis and Structural Characterization of U−ea and U−ea/sb

First, the U−ea monomer was synthesized by the Mannich condensation using natural urushiol and ethanolamine as the phenolic and amine sources, respectively (Figure 2). The Appendix A shows the synthesis and characterization of U−ea. Subsequently, the U−ea/sb monomer was synthesized by the reaction of a tertiary amine on the U−ea oxazine ring with 1,3−propanesultone. The chemical structures of as-prepared U−ea and U−ea/sb monomers were characterized by ATR-FTIR spectroscopy. Figure 1a shows the ATR-FTIR spectra of urushiol (U), U−ea, and U−ea/sb, as well as the analyses of the characteristic peaks of the major functional groups. In the ATR-FTIR spectra of U, U−ea, and U−ea/sb, the characteristic absorption peaks at 981 cm^−1^ and 945 cm^−1^ corresponded to the bending vibrations of the triene structure (-CH=CH-CH_2_-CH=CH-CH=CH-), as shown in Figure 1a; these vibrations were characteristic of the long side chains of urushiol, and the peaks at 2853 cm^−1^ and 2924 cm^−1^ corresponded to the stretching vibrations of the saturated carbon-carbon bond (Figure 1a). Compared with the ATR-FTIR spectrum of U, those of U−ea and U−ea/sb revealed a new characteristic absorption peak at 963 cm^−1^ (Figure 1a), corresponding to the C-H out-of-plane bending vibrations on the oxazine ring, confirming the successful synthesis of the urushiol-based benzoxazine monomers in the previous stage; this result is in agreement with the ^1^H NMR test results. Moreover, the characteristic absorption peak of the oxazine ring was observed in the ATR-FTIR spectrum of U−ea/sb, indicating that the introduction of sulfobetaine did not destroy the oxazine ring structure and that functionalized benzoxazine was successfully synthesized. In addition, in the ATR-FTIR spectrum of U−ea/sb, distinct single peaks were observed at 1164 cm^−1^ and 1034 cm^−1^, corresponding to the stretching vibrations of the−SO_3_^−^ group, indicative of the formation of sulfobetaine groups (Figure 1a). In the ATR-FTIR spectrum of urushiol, the broad absorption band at 3349 cm^−1^ corresponded to the vibrations of hydroxyl groups; the absorption band at 3009 cm^−1^ corresponded to the stretching vibrations of isolated =C–H groups. In the ATR-FTIR spectra of U−ea and U−ea/sb, a stretching vibration peak of isolated =C–H groups was also observed at 3009 cm^−1^. Thus, the syntheses of U−ea and U−ea/sb did not alter the isolated =C–H groups of the long side chain. The ATR-FTIR spectra of the coated poly(U−ea) and poly(U−ea/sb) (Figure 1b) were compared with those of U−ea and U−ea/sb (Figure 1a): Clearly, significant differences were observed. The characteristic peaks of the oxazine ring at 963 cm^−1^ and the absorption peaks of the triene structure in the long side chains at 981 cm^−1^ and 945 cm^−1^ were not clearly observed in the ATR-FTIR spectra of poly(U−ea) and poly(U−ea/sb) (Figure 1b). In addition, the stretching vibrations of the isolated =C–H groups at 3009 cm^−1^ disappeared. Moreover, the two stretching vibrational peaks of the −−SO_3_^−^ group were still observed in the ATR-FTIR spectrum of poly(U−ea/sb) (Figure 1b). Differences in the ATR-FTIR spectra of the polymers and monomers suggested that the ring-opening polymerization (ROP) of benzoxazine occurred to form bulk structures at a temperature of greater than 140 °C. Thermal ROP is a typical polymerization reaction of the benzoxazine monomer. At the start of ROP, the oxazine ring forms C positive ions and iminium ions. With the increase in the temperature, the electrophilic substitution reaction of the C positive ions is favored for chain growth, which in turn accelerates the rate of ROP and finally forms a polymer with a high molecular weight; hence, the absorption peaks corresponding to the oxazine ring skeleton in the ATR-FTIR spectra disappeared. The difference between the ATR-FTIR spectra of poly(U−ea) and poly(U−ea/sb) revealed that heating curing did not destroy the sulfobetaine structure of poly(U−ea/sb) and that the presence of quaternary ammonium salts rendered excellent antifouling properties to the poly(U−ea/sb) coating.

### 3.2. Surface Chemical Composition Analysis of Poly(U−ea/sb)

XPS survey spectra were recorded to analyze the surface chemical compositions of poly(U−ea) and the poly(U−ea/sb) coatings. The presence of the unique quaternary (R_4_N^+^) and sulfonic acid (−−SO_3_^−^) groups of the poly(U−ea/sb) coatings was confirmed by the high-resolution XPS spectra of the N1s and S2p regions. As shown in Figure 2a, the comparison of the poly(U−ea) and poly(U−ea/sb) XPS spectra revealed the presence of only three peaks of C1s, N1s, and O1s, respectively, in the full XPS spectrum of poly(U−ea); however, two new peaks corresponding to S were clearly observed in the XPS survey spectrum of the poly(U−ea/sb) coating, which was synthesized by the introduction of sulfobetaine groups into the polybenzoxazine chains. Furthermore, according to the peak fitting process of XPS high-resolution spectra, the main spectral bands were assigned according to the binding energy (BE), as shown in Figure 2b–d. The analysis of the N1s region revealed slight broadening and two distinct nitrogen peaks for the poly(U−ea/sb) coating (Figure 2c) compared to the one main peak observed for the poly(U−ea)-coated surface (Figure 2b), where the main peak (~400.83 eV) corresponded to the Ar-CH_2_-N bond, while the other peak (~402.22 eV) provided evidence for the formation of quaternary ammonium groups (^+^NR_4_). As the quaternary ammonium peak was electron deficient and more stable, the quaternary peak transferred higher energy to displace electrons from its N1s orbital. The integral ratio revealed the presence of 7.6 mol% of the ^+^NR_4_ fraction in the poly(U−ea/sb) coating relative to the nitrogen content of the oxazine ring Ar-CH_2_-N. The grafting rate of sulfobetaine was 28.18% based on the area ratio of the two nitrogen elements. In addition, the core-level S2p spectra of the poly(U−ea/sb) coating revealed two peaks at 167.7 and 168.8 eV, corresponding to the S2p,1/2 and S2p,3/2 signals of the SO_3_^−^ groups, respectively (Figure 2d). These results revealed that amphoteric benzoxazine antifouling coatings containing both cations (^+^NR_4_) and anions (SO_3_^−^) were successfully prepared by the reaction of 1,3−propanesultone with tertiary amines.

### 3.3. Properties of Poly(U−ea/sb)

As the U−ea/sb monomer structure still comprised benzoxazine groups, the poly(U−ea/sb) coating was viewed as a polybenzoxazine coating containing zwitterionic moieties (sulfobetaine). Figure 3a shows the DSC thermograms of poly(U−ea) and the poly(U−ea/sb) coatings. The ROP of benzoxazine is accompanied by an exothermic reaction; however, the poly(U−ea) and poly(U−ea/sb) coatings did not exhibit exothermic peaks in the DSC spectra during the second heating scan, confirming that the benzoxazine ring was opened during the thermal curing of U−ea and U−ea/sb into coatings and that the introduced sulfobetaine group did not affect the ROP of the oxazine ring. Compared to the tertiary amine of poly(U−ea), the quaternary ammonium salt of poly(U−ea/sb) possibly increased the ring strain of the benzoxazine moiety and weakened the bond strength of the adjacent CH_2_-O bond, indicating that grafting sulfobetaine into the benzoxazine ring promoted its ring opening. In the DSC thermograms, the T_g_ values of the poly(U−ea/sb) coating were slightly less than those of the poly(U−ea) coating, possibly related to the increase in the mobile fragments of the poly(U−ea/sb) coating. Sulfobetaine provided spatial site resistance to the crosslinked network, resulting in an increase in the free fragment volume fraction and a decrease in the crosslinking density of poly(U−ea/sb).

The thermal stability of the poly(U−ea/sb) coating was investigated by TGA (based on the initial temperature, that is, the degradation temperature at 5% weight loss). As shown in Figure 3b,c, the poly(U−ea/sb) coating started to decompose at 205 °C, and the test ended with a residual carbon content of 18.0 wt%. Clearly, the structure resulting from the complete ROP of the U−ea/sb monomer and the cross-linking of the reactive sites of the benzene ring side chains allowed the introduction of a more stable sulphobetaine moiety, indicative of better thermal stability.

Static WCA is one of the key indicators to evaluate the hydrophilicity of the material surface. Figure 3d shows the results obtained. As zwitterionic polymers comprise the same cationic and anionic group units, they can react with water molecules via electrostatic interactions, forming a strong hydration layer to enhance hydrophilicity. Therefore, the change in the WCA can further confirm the successful preparation of the sulfobetaine-functionalized benzoxazine coatings. The WCA of the poly(U−ea) coating was 95.2° ± 1.9°, indicative of significant hydrophobicity; this result was in agreement with the low surface energy and strong hydrophobicity of the polybenzoxazine coating. The inherent hydrophobicity of polybenzoxazine was attributed to the crosslinking of unsaturated aliphatic chains on the benzene ring. With the increase in the crosslinking density of the long side chains, the resistance of the polar phenolic hydroxyl groups to interact with water molecules also increased. Compared with that of the control benzoxazine coating poly(U−ea), the hydrophilicity of the poly(U−ea/sb) coating was significantly higher, and the CA decreased to 37.6° ± 4.1°, corresponding to a decrease of 60.52%. The CA test results indicated that the hydrophilicity of the coating was considerably improved because of the successful grafting of sulfobetaine onto the oxazine ring. When water drops were dropped onto the poly(U−ea/sb) film surface, the water drops spread rapidly. This phenomenon was attributed to the high surface tension of the poly(U−ea/sb) coating, which enhanced the interaction between sulfobetaine and water molecules. The CA test results revealed that the poly(U−ea) layer was hydrophobic and exhibited low adhesion to water; in contrast, owing to the abundance of cations and anions in the poly(U−ea/sb) coating, it exhibited hydrophilicity and high adhesion to water. The results obtained from the water adhesion tests of poly(U−ea) and the poly(U−ea/sb) coatings shown in Figure 3e provided strong evidence. Figure 3f shows the results of adhesion force statistics. Clearly, the test results revealed that compared to the poly(U−ea) coating, the poly(U−ea/sb) coating exhibited a higher surface adhesion to water and that the average adhesion force of the poly(U−ea/sb) coating to water was 0.57 mN, while that of the poly(U−ea) coating was only 0.44 mN. This test result was in agreement with the CA test results.

### 3.4. Mechanical Properties of the Poly(U−ea/sb) Coatings

Ships sailing in the ocean are constantly subjected to the harsh effects of seawater on their hull surfaces, making it imperative for coatings to exhibit strong mechanical properties. As shown in Table 1, the hardness of the poly(U−ea) and poly(U−ea/sb) coated on tinplate sheets are 8H and 5H, and the adhesion strength to the substrate are 0.28 MPa and 0.56 MPa, respectively. The glass transition temperature of poly(U−ea/sb) was slightly lower than that of poly(U−ea) due to the decrease in crosslinking density and increased distance between molecular segments (Figure 3), which resulted in a reduction in hardness and brittleness. Zwitterion promotes the ring-opening addition of the oxazine ring, in which active methylene reacts strongly with electrophilic groups on the substrate, resulting the adhesion of the coating to the substrate was significantly improved after amphoteric ionization. In addition, as shown in Figure 4 tensile tests were carried out on both coatings, with poly(U−ea/sb) showing higher tensile strength (11.84 MPa) and elongation at break (6.45%) than poly(U−ea) (4.41 MPa and 0.59%, respectively). The corresponding toughness and elastic modulus of the poly(U−ea/sb) and poly(U−ea) coatings were 49.25 MJm^−3^ and 378.51 MPa and 1.16 MJm^−3^ and 580.41 MPa, respectively. These results revealed that the addition of sulfobetaine significantly reduced the elastic modulus and increased the toughness of the poly(U−ea/sb) coating. In summary, the addition of sulfobetaine exerts some positive effects on substrate adhesion and the mechanical properties of the coating.

### 3.5. Antiprotein Adsorption Performance of the Poly(U−ea/sb) Coatings

Protein adhesion is fundamental in several relevant biological reactions, such as biofilm formation and cell adhesion; in addition, it is well correlated to the antifouling properties of membranes. As bacteria can secrete a high number of proteins, including accumulation-associated proteins (*App*) and fibronectin-binding proteins (*FnBps*), bacteria are firmly adhered to the material surface by these proteins. Therefore, the static adhesion of proteins is the main method to assess the antibiofouling performance of the coatings. Test methods for coating resistance to protein adhesion are in the Appendix A and all tests are performed under lab conditions. Two major plasma proteins were selected for testing: *BSA* and *γ-globulin*. Moreover, BG and the poly(U−ea) coatings were used as the control samples. Protein adhesion was evaluated by immersing the membranes in a 2 mg/mL *BSA* or *γ-globulin* solution, and the amount of the adhered protein was estimated by measuring the concentration of the protein adhered on the sample surface. Proteins can be directly semi-quantified at 280 nm to compare different concentrations using UV-visible (*UV-vis*) spectroscopy because 280 nm is the maximum absorption value typically observed for proteins containing tryptophan, tyrosine, and cystine (disulfide-bonded cysteine residues). Albumin contains 17 disulfide bonds and thiol groups, comprising free cysteine (*Cys*) [30,31]. Figure 5a and b show the UV absorption values at 280 nm for the *BSA* and *γ-globulin* solutions on different sample surfaces [32]. Clearly, the Abs values of both protein solutions on the poly(U−ea/sb) sample surface were less than those of BG and poly(U−ea) (Figure 5); that is, the adhesion of *BSA* and *γ-globulin* on the poly(U−ea/sb) sample surface was less than that of the control. Owing to the hydrophilic properties of the zwitterionic polymers, the hydrophilic groups on the poly(U−ea/sb) coating surface adsorbed water molecules on the surface via hydrogen-bonding interactions, forming a hydrated layer; this layer hindered the interaction between the proteins and surface, thus preventing protein adhesion. As can be clearly observed in Figure 5c, the UV absorbance of the *BSA* solution on the poly(U−ea/sb) sample surface was 0.32, which was greater than that of *γ-globulin*. The results demonstrated that the oxazine ring of benzoxazine was successfully grafted with zwitterionic sulfobetaine and that the adhesion of proteins on the membrane surface was considerably reduced because of the action of zwitterionic sulfobetaine.

### 3.6. Antibacterial Performance of Poly(U−ea/sb) Coatings

In contrast to protein adhesion, bacterial adhesion on the hull surface may adversely affect the performance and lifetime of the hull. To examine the antibacterial performance of the poly(U−ea/sb) coating, four typical bacteria were used: Gram negative bacteria, viz. *E. coli* (*BW 25113*) and *V. alginolyticus* (*ATCC 33787*) and Gram positive bacteria *S. aureus* (*ATCC 25923*) and *Bacillus.* sp. (*MCCC 1B00342*). The specific test procedures for antibacterial and antibacterial adhesion testing of the coating are in the Appendix A. In addition, the ability to kill bacteria was quantified by calculating the number of bacterial colonies on agar plates from different sample isolates relative to the number of bacterial colonies on the BG samples to estimate the antibacterial rate [33]. As shown in Figure 6, a large number of bacterial colonies were observed on the agar plates incubated with the bacterial isolates from BG and poly(U−ea), while almost no colonies were observed on the agar plates incubated with the bacterial isolates from the poly(U−ea/sb) coating. A 20% inhibition rate for the four bacteria on the poly(U−ea) coating was observed, corresponding to a poor inhibition effect, while the inhibition rate of the poly(U−ea/sb) coating was greater than 99.99%, indicative of a good inhibition effect. The high inhibition rate of the poly(U−ea/sb) coating was mainly attributed to the structure of the quaternary ammonium salt in the betaine structure. The cation N^+^ made the poly(U−ea/sb) coating surface positively charged, which combined with the negative charge in the phospholipid bilayer on the bacterial cell membrane via electrostatic interactions, leading to the destruction of the cell membrane and the outflow of DNA, proteins, and other substances from the cell and subsequently leading to cell necrosis. Regardless of the type of bacteria, i.e., Gram positive or Gram negative bacteria, the cell surface was typically negatively charged. The antibacterial test results revealed that the poly(U−ea/sb) coating exhibited excellent antibacterial properties and can effectively kill the bacteria adhering to the coating surface. To further determine the extent of biofilm formation on the poly(U−ea) and poly(U−ea/sb) coating surfaces, FE−SEM images were recorded to visualize the adhesion of bacteria on the poly(U−ea) and poly(U−ea/sb) coating surfaces (test under lab conditions). Figure 7 shows the SEM images of *E. coli*, *S. aureus*, *V. alginolyticus*, and *Bacillus.* sp. cultured on different coatings. A number of bacteria were visible on the BG surface, while fewer bacteria adhered to the poly(U−ea) and poly(U−ea/sb) surfaces. From the results of the CA test and droplet adhesion test in Section 3.3, the poly(U−ea) coating exhibited a low surface energy; in general, low−Surface-energy coatings exhibited the least adhesion to fouling. In addition, the SEM image of poly(U−ea) revealed a smooth, flat, and dense surface, which also helped to reduce the adhesion to bacteria. Owing to the presence of abundant anions and cations on the poly(U−ea/sb) coating surface, it was extremely easy to form a water layer on the surface to hinder bacterial adhesion, while in the presence of quaternary ammonium salts (N^+^), the bacteria adsorbed on the membrane surface were killed and easily stripped by hydraulic turbulence [34].

### 3.7. Algal Inhibition Performance of the Poly(U−ea/sb) Coatings

To further investigate the antifouling performance of the poly(U−ea/sb) coating, three common marine microalgae, i.e., *N. closterium*, *P. tricornutum*, and *D. zhanjiangensis*, respectively, were selected for the investigation of algal contamination, algal growth, and attachment experiments (test under lab conditions). The Appendix A revealed the procedure to evaluate the algae inhibition performance. First, the inhibitory effect of the poly(U−ea/sb) coating on *N. closterium* was examined. The sterilized BG, poly(U−ea), and poly(U−ea/sb) coatings were immersed in the medium of f/2 of diluted algal cells *N. closterium* for 1 week, the concentration of *N. closterium* cells was determined by counting the number of cells using a blood cell counter (Refine Biochemical Reagent Instrument Co., Ltd., Shanghai, China), and optical photographs of the algal growth process were recorded (Figure 8a). After 1 day of immersion, the concentration of *N. closterium* was less in the early growth stage, and no significant difference between the three coatings of BG, poly(U−ea), and poly(U−ea/sb) was observed. After 7 days of immersion, the inhibitory effects of the BG, poly(U−ea), and poly(U−ea/sb) coatings on *N. closterium* were significantly different. The surfaces of BG and poly(U−ea), as well as the bottom of the culture medium, were accompanied by a high number of yellow-green precipitates, indicative of considerable proliferation of algal cells, while the solution in the culture dishes of poly(U−ea/sb) plates was clear and transparent, with no clear proliferation of algal cells. Furthermore, the poly(U−ea/sb) coating did not exhibit significant swelling, and the coating was intact overall. The concentrations of *N. closterium* cells in the BG and the poly(U−ea)-coated glass dishes at this time were 112.96 (±27.97) × 10^5^ cell/mL and 110.38 (±24.37) × 10^5^ cell/mL, respectively. In contrast, the concentration of algal cells in the poly(U−ea/sb)-coated dishes was only 1.63 (±0.18) × 10^5^ cell/mL, with no significant change compared to the initial algal cell concentration (Figure 8d).

In this study, the inhibitory effect of the poly(U−ea/sb) coating on different algal species (*P. tricornutum* and *D. zhanjiangensis*) was also investigated. *P. tricornutum* and *D. zhanjiangensis* algal cell solutions were cultured in f/2 medium with the BG, poly(U−ea), and poly(U−ea/sb) coatings (Figure 8b,c). After 7 days of culture, a high number of *P. tricornutum* and *D. zhanjiangensis* cells proliferated in the medium containing BG and poly(U−ea), exhibiting extremely high growth activity. In contrast, the growth activity of algal cells was extremely low in the medium with *P. tricornutum* and *D. zhanjiangensis* with cell concentrations of 2.9 (±1.9) × 10^5^ cell/mL and 1.63 (±0.53) × 10^5^ cell/mL in the medium with the poly(U−ea/sb) coating, respectively. In contrast, the cell concentrations of *P. tricornutum* and *D. zhanjiangensis* in the BG-containing medium were 112.25 (±22.27) × 10^5^ cell/mL and 30.64 (±1.94) × 10^5^ cell/mL, respectively. The cell concentrations of *P. tricornutum* and *D. zhanjiangensis* in the poly(U−ea)-coated medium were 135.1 (±12.2) × 10^5^ cell/mL and 41.0 (±2.0) × 10^5^ cell/mL, respectively. The results confirmed that BG and poly(U−ea) did not inhibit the growth of algal cells and that algal cells were easily attached to the surface, while the poly(U−ea) coating exhibited an algal inhibition rate of >99%. The results of the 7 days algae inhibition experiment revealed that the antifouling properties of the poly(U−ea/sb) coating could last for a long time, which is crucial for practical applications. The plates of BG, poly(U−ea), and poly(U−ea/sb) were removed from the test medium after 7 days of soaking and rinsed with 20 mL of sterile PBS to wash off non-adherent algae. Subsequently, the three plates were examined for algal adherence using a fluorescence microscope (Zeiss Axio Imager A2, Oberkochen, Germany), and images of five random areas (40× magnification, 0.156 mm^2^/area) of each sample were recorded. The algal coverage on the BG, poly(U−ea), and poly(U−ea/sb) plates was determined by analyzing fluorescence microscopy images using ImageJ software. All results and standard deviations are based on triplicate experiments. The optical photographs of the plates revealed the adherence of a large number of algae to the BG and poly(U−ea) plates, while only a small number of algae adhered to the poly(U−ea/sb) plates (Figure 9a–c). The densities of the algal cell coverage on the surfaces of *N. closterium*, *P. tricornutum*, and *D. zhanjiangensis* in the BG medium were 17.44 ± 2.14%, 19.91 ± 1.44%, and 15.12 ± 1.13%, respectively (Figure 9d). However, the density of algal cells on the poly(U−ea/sb) coating surface decreased to almost 0%. This phenomenon can be attributed to two reasons. First, the anions and cations on the poly(U−ea/sb) surface act as a physical and energetic barrier via ionic solubilization, and the water molecules in the hydrated layer are tightly bound together, preventing the adsorption of the algal cells to the surface. Second, because of the quaternary ammonium salt structure, the poly(U−ea/sb) coating inhibits the reproduction of a large number of algal cells, which can kill the algae adhering to the coating for reducing the number of algal cells adhering to the coating surface.

## 4. Conclusions

In summary, a benzoxazine precursor was synthesized by the Mannich condensation, which was reacted with 1,3−propanesultone to successfully prepare a poly(U−ea/sb) antifouling coating with dual “offensive and defensive” antifouling properties. Ex-periments revealed that the poly(U−ea/sb) coating not only efficiently killed the attached bacteria and algae but also exhibited a good inhibition effect for the adhesion of non−Specific proteins. By the attack of fouling organisms, the “active offensive” and “passive defensive” mechanisms were activated simultaneously, rendering an efficient and long-lasting antifouling performance to the coating (Figure 3). The use of the benzoxazine precursor rendered excellent heat resistance and long-term stability to the coating, while the poly(U−ea/sb) coating surface was smooth, dense, and non-porous as the poly(U−ea) coating surface. As a green, biodegradable coating, poly(U−ea/sb) was composed of the natural product urushiol, with non-polluting quaternary ammonium salts instead of traditional antifouling agents for antibacterial and algae inhibition, while simultaneously relying on a strong hydrophilic surface to form a hydrated layer that physically blocked the adhesion of non-specific proteins. As a marine antifouling coating, the poly(U−ea/sb) coating not only exhibited 99% antibacterial and antialgal rates, but also the water contact angle was reduced by 60.52% compared with that of the poly(U−ea) coating, and the hydrophilicity was considerably improved, which exhibited better resistance to the adhesion of foreign proteins and adhesion of bacteria. Therefore, considering its high efficiency and environmental friendliness, the poly(U−ea/sb) antifouling material demonstrates potential to solve the economic loss caused by marine biofouling and environmental pollution related to the abuse of antifouling agents.

## Data Availability

The data presented in this study are available on request from the corresponding author.

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
