# Peer review of "Urushiol-Based Benzoxazine Containing Sulfobetaine Groups for Sustainable Marine Antifouling Applications"

_polymers, 2023, doi:10.3390/polym15102383_

Round 1

Reviewer 1 Report (New Reviewer)

This manuscript describes the urushiol-based Benzoxazine Containing Sulfobetaine Groups coatings for Marine Antifouling Applications. The topic is of scientific interest due to the growing need for sustainable ecofriendly antifouling applications. Though the manuscript is prepared well, it looks like incomplete. The methodology part for protein adhesion, antibacterial and antialgal assays are missing. I am not able assess any supplementary files for those methods. As the methodology is not available, it is not clear about the controls used for the experiments. I would like to suggest following points for revision

1. The introduction section is containing too many sentences that are common in scientific domain. Try to concise the introduction part

2. Provide the methodology for antifouling assays

3. Analyse the data using appropriate statistical tools

4. Whether the adhesion ability of these bacterial strains are tested under lab conditions. Clarify this in the methodology. 

Though I am not qualified to judge the language, I feel that moderate editing required

Author Response

Reviewer 2 Report (New Reviewer)

See the attached.

Round 2

Reviewer 1 Report (New Reviewer)

The authors have improved the manuscript with additional inputs. 

This manuscript is a resubmission of an earlier submission. The following is a list of the peer review reports and author responses from that submission.

Round 1

Reviewer 1 Report

Here are a few comments for improvement

ü  Please check the referencing styles. Please end the sentences including the references.

ü  Please take off unnecessary references. e.g., L42 – 7-10, you need only one or two sentences to support your statement. L44 same. Please make changes wherever necessary.

ü   L42 – change ‘Long’ to long.

ü  L49 – English- check full stops

ü  L54- add full stop after reference

From hereafter no comments on general English. This paper requires a professional proofreading

ü  The introduction needs to be restructured.

ü  Introduction should maintain the flow and reflect well the story (novelty) of the research. Scheme 1 should be under the materials and methods

ü  2 and 2.1 under the same heading.- please change the heading

ü  3.3 why this heading?

ü  Please justify why you have carried out these characterisations in one or two sentences in the introduction.

ü  L302 – 305 please justify the statement or re-write to reflect better what the findings are.

ü  L349 – please change highly hydrophobic to hydrophobic.

ü  How accurate is the measurement if the water spreads quickly?

ü  Please justify how the new coating can be called “green” / environmental friendliness – is this coating greener or more environmentally friendly?

 Many thanks,

Reviewer 2 Report

The paper by Xu, Lin et al. describes an urushiol-based benzoxazine resin suggested for marine application. In my opinion, the study is conceptionally wrong. The authors tried to synthesize high-performance durable coating that is biodegradable, which is in contradiction. In principle, the biodegradable coating should work well for indoor applications but not for marine application - where degradation by environmental conditions should go on. Furthermore, sources of urushiol are very limited for such market.

Further major points:

-          NMR spectrum reposted for the benzoxazine resin is hardly acceptable. It does not prove successful synthesis of the resin.

-          In IR spectra, I am missing discussion related with unsaturated lipidic tails. They should give a band at ~3010 cm-1. Did you observe it in urushiol? Does it changed upon synthesis?

-          I am surprised that the curing at elevated temperature did not touch the unsaturated lipidic tails (I found not discussion of this phenomenon). Do you have some explanation?

-          I am missing recent literature on benzoxazine resins/coatings (e.g., Green. Chem. 2020, 22, 1209; Prog. Org. Coat. 2023, 174, 107298.

-          Missing Tg values of to the coatings.

-          Missing some data comparing for comparison with commercial marine coating. How could you judge about “high-performance”?

-          Adhesion force is hard to compare with literature data. Adhesion strength (in MPa) should be calculated, as it is more appropriate parameter.

-          Missing stress-strain behavior of the materials and DTA analysis. These data are very important and can help the comparison with literature benzoxazine materials.

In summary, I cannot recommend the paper for publication in present form.